# Metric for Structural Complexity Assessment of Space Systems Modeled Using the System Modeling Language

**Victor Emmanuel Pierre Lopez** [1],*[iD] **and Lawrence Dale Thomas** [2][iD]

1 Mechanical and Aerospace Engineering Department, University of Alabama in Huntsville, Huntsville, AL 35899, USA
2 Industrial & Systems Engineering and Engineering Management Department, University of Alabama in Huntsville, Huntsville, AL 35899, USA
* Correspondence: victor.lopez@uah.edu

**Abstract:** A complexity metric is proposed for the quantification of system complexity using information about the composition of a system and its interactions depicted in a System Modelling Language (SysML) model. The proposed metric is adapted from the complexity metric developed for design structure matrix (DSM) applications and was modified to allow the metric to be applied at different decomposition levels and to accommodate the inclusion of external interactions. The metric was applied to three case studies: a Mars lander, a CubeSat and a spacecraft thermal control system. The proposed metric attributed a higher amount of complexity due to the interactions compared to the DSM metric. This variance resulted in instances where the results differed for the two metrics. Despite these differences, both metrics behaved similarly to changes in component or interaction complexity.

**Keywords:** complexity metric; SysML model; model architecture; design effort





## 1. Introduction

There is an ever-increasing demand for space system capabilities, systems need to do more, perform better and in new and more diverse environments. This increase in capability results in an increase in system complexity [1]. Higher system complexity is indicative of a higher difficulty to develop and manage a system which can fail in ways that are hard to predict [1,2]. Due to the trend of increased system complexity, the ability for system designers to assess and manage the complexity of their design throughout the life cycle will become a critical tool for successful system realization and sustainment.

Although there is a large and ever-growing amount of literature about complexity, there are no established methods for every day systems designers to quantify the complexity of their design directly with the information already available to them. This paper proposes a metric to quantify system complexity based on information usually captured while practicing model based systems engineering (MBSE). The objective of the metric is to provide system designers and modelers a method to easily quantify and manage the complexity of their design. The metric's viability will be evaluated by verifying Weyuker's properties and applying the metric to different systems for comparison to other similar established work.

## 2. Complexity for Engineering Design

### 2.1. Complexity Definition

Complexity has a wide array of definitions and measures depending on the context and use of the complexity concept. This work will reference the concept of logical depth as the basis for the understanding of complexity [3]. Given the algorithmic complexity of an object, logical depth is the time spent to run the algorithm to recreate the original description of the object. The longer the time, the 'deeper' or more complex the description and the object are considered [2,3].

A parallel will be used between system complexity and the logical depth of an object, where the algorithmic complexity is replaced by the system specifications, and the algorithm to recreate the original object is replaced by the set of processes required for system development; additionally, the notion of time will be expanded to be the set of resources used during the system development. As such system complexity will be defined as 'the amount of resources spent to develop or acquire a working system based on the system specifications', where the system specifications are understood to be the set of information describing the system used during development such as requirement tables, flow diagrams, connection diagrams, and structural decompositions that describe the system and the mission it needs to accomplish. The definition is broad enough to encompass many different aspects of complexity while basing complexity on real world aspects of the system development.With the overall definition of complexity established, a metric will be proposed to quantify the structural complexity of a system where the structural complexity is defined as the complexity due to the logical and physical representation of the structure of system.

### 2.2. Proposed Complexity Metric

The metric shall rely on elements commonly accepted to contribute to structural complexity mentioned in the literature: (i) The size or number of parts of a system (ii) the interactions between the parts and (iii) the intricacy of the interactions or lack of modularity of the design. To follow the engineering and contractual practice of decomposition, the metric will also need to be capable of being applied at different decomposition levels such as the system, subsystem, and subcomponent level as needed. Other aspects commonly associated with complexity such as behavioral descriptions of the non-linearity or intricateness of a behavior [2,4] will be marked as future work to be investigated in another study.

#### 2.2.1. Leveraging Model Based Systems Engineering (MBSE)

Historical methods for structural complexity estimation require effort and time to gather and analyze the specifications of a system to represent them in a format that is viable for the complexity estimation method such as DSM [5–9], a graph [10,11], or a description string [2,12]. These specifications have historically been organized in a set of documents that are not machine readable. The implementation of MBSE in a project moves most of the specifications to be stored within a systems model, usually written using the Systems Modelling Language (SysML) or a derivative of SysML (e.g., Unified Architecture Framework Profile (UAFP)) for different tasks such as conceptual design, requirement verification, trade studies, technical analysis, and other domain specific tasks. The implementation of MBSE usually will often involve the creation of a structural representation representing the system composition and its interactions; This information is usually depicted using two types of SysML diagrams – Block Definition Diagrams (BDDs) and Internal Block Diagrams (IBDs). The BDD is a common way to depict the general structure of a system and its organization, and it can depict the structural decomposition of the system to different levels and the relations to other structural elements. The IBD shows the interactions between internal components. Methodologies exist for the creation of these models in a systemic way [13,14]

The existence and use of these SysML models provides an opportunity for complexity estimation since these models are a centralized and machine readable source of information. As such the complexity metric proposed will leverage the information depicted in these representations as the basis for complexity estimation.

2.2.2. Complexity Metric Definition

A metric developed by Sinha and Suh was identified as a candidate to measure the complexity of a design. The metric requires a Design Structured Matrix (DSM) representation of the system [7]. The complexity metric is defined as

$$C_{DSM,k} = \sum_{i=1}^{n} \alpha_i + \left( \sum_{i=1}^{n} \sum_{j=1}^{n} \beta_{ij} A_{ij} \right) \frac{E(A)}{n} \tag{1}$$

where $C_{DSM,k}$ is the complexity of the system of interest $k$, $\alpha_i$ is the complexity of the $i$th subcomponent, $\beta_{ij}$ is the complexity of the interaction between the $i$th and $j$th subcomponents, $A_{ij}$ is the adjacency matrix, $E(A)$ is the graph energy of the adjacency matrix, and $n$ is the number of subcomponents. This metric will be hereafter referenced as the "DSM complexity metric".

Further study into the DSM complexity metric revealed two problems related to the use of the adjacency matrix. As mentioned previously one of the requirements is for the metric to be applied at different hierarchical levels. The use of an adjacency matrix requires a N × N representation of the system; this results in only being able to apply the metric to the whole system at the highest hierarchical level or to lower levels but neglecting external interactions. In addition, the adjacency matrix also renders difficult the study of situations where the information about components is not equal across hierarchical levels, e.g., when assessing commercial of the shelf (COTS) solutions alongside in-house components. The second problem stems from the use of graph energy as the measure of topological complexity since its calculation is also dependent on an adjacency matrix.

To apply the metric at different decomposition levels, leveraging both internal and external information found in SysML models, the DSM complexity metric will be adapted. To avoid the use of an adjacency matrix, a replacement for graph energy as a measure connection density is needed. Graph energy as shown in Equation (1) is used to quantify the topological complexity due to the connectivity of the different elements and related to the effort of integration [7]. Higher connection density is indicative of a design that has a higher number of design dependencies which in turn result in consequences such as changes that propagate through those paths [15]. Cyclomatic complexity as adapted by Lankford [10] for class diagrams, shown in Equation (2), is intended to show the intricacy of the interactions between classes; Given that it is easily measurable using SysML models; cyclomatic complexity is proposed as a replacement for the graph energy $E(A)$, where cyclomatic complexity is defined as

$$v = e - n + 2 \cdot p \tag{2}$$

where $v$ is the cyclomatic complexity of the intended block, $e$ is the number of relations and $p$ is the number of connected internal components. The resultant modification to Equation (1) yields the following equation

$$C_{DSM,k} = \sum_{i=1}^{n} \alpha_i + \left( \sum_{i=1}^{n} \sum_{j=1}^{n} \beta_{ij} A_{ij} \right) \frac{v}{n} \tag{3}$$

The term that accounts for the complexity due to the interactions, $\sum \sum \beta_{ij} A_{ij}$ will also need to be modified to be the sum of the complexity of each interaction due to the potential existence of external interactions. Since SysML models typically depict each interaction directly, this change allows data characterizing particular interactions to be leveraged. This also translates to a new term added to take into account the complexity of the external interactions.

With the previous adaptations made the resulting metric that is proposed is defined as

$$C_{SCM,k} = \sum_{i=1}^{n} \alpha_i + \left( \sum_{i=1}^{n} \sum_{j=1}^{n} \beta_{ij} + \sum_{i=1}^{n} \sum_{m=1}^{p} \gamma_{im} \right) \frac{v}{n} \tag{4}$$

where $C_{SCM,k}$ is the complexity of the component of interest $k$, $\alpha_i$ is the complexity of the $i$th subcomponent of $k$, $\beta_i$ is the complexity of the interaction between the $i$th and $j$th subcomponents of $k$, $\gamma_{im}$ is the complexity of the interaction between the $i$th subcomponent of $k$ and the mth external subcomponent, $v$ is the cyclomatic complexity of $k$, $n$ is the number of subcomponents, and $p$ the number of external subcomponents.

$\alpha_i$ can be determined either by expert opinion of the resources expended to acquire the $i$th subcomponent or by applying Equation (4) to the subparts comprising the $i$th subcomponent. $\beta_{ij}$ and $\gamma_{im}$ determined through expert opinion based on the resources that would be spent to make the interaction successful, or by another technique similar to the one used by the DSM Complexity metric [8], assuming the complexity of the interaction to be a percentage of the complexity of the component. For simplicity, the metric defined in Equation (4) will be hereafter referenced as the "SysML complexity metric".

The definition of complexity allows for future verification of the metric based on quantitative data from system developments such as cost, schedule, and engineering effort. An example of how the metric can be applied for a simple distiller be found in an earlier work [16].

A summary of the differences between the DSM and SysML complexity metrics can be seen in Table 1. It can be observed that both metrics assess complexity based on the structural representation of the system. The main difference between the two metrics resides in the goal of each metric and the representation of the system used. Indeed, the DSM complexity metric would be more appropriate in situations where the representation of the system is easier as a DSM, and the goal is to assess the overall system complexity as it relates to system modularity.

**Table 1.** Summary comparison between SysML complexity metric and DSM complexity metric [7].

| Metric | SysML Complexity Metric | DSM Complexity Metric |
|---|---|---|
| Basis of Metric | SysML Model | DSM |
| Knowledge of system required | SysML description of the components and interactions up to level for which complexity can be confidently assessed. Mixing levels of decomposition is allowed. External interactions can be included | DSM showing interactions between all components at the same decomposition level with associated complexities for components and interactions. External interactions cannot be included |
| Goal | Quantify complexity throughout the design, to provide system designers insight into the complexity of their design, the origin of the complexity, and the potential consequences of said complexity such as unexpected behavior or larger resources needs | Quantify overall system complexity to provide system designers an understanding of the system complexity and the best arrangement of subsystems to increase modularity while minimizing complexity |
| Calculation Procedure | From a SysML model, it can be calculated directly from the SysML tool using constraints relationships or with an external tool using an XML export of the model. | From a DSM description, it can be easily calculated using matrix math using Excel or MATLAB. |
| Interaction representation | SysML Model element interactions | Adjacency Matrix |
| Topoligcal Complexity term | Cyclomatic Complexity | Graph Energy |

*2.3. SysML Complexity Metric Discussion*

2.3.1. Verification of Weyuker's Properties

1. $(\exists P)(\exists Q)(|P| \neq |Q|)$ where $P$ and $Q$ are two different classes. A measure should not rank all classes as equally complex. *Proof*: Consider element $A$ composed of elements $B$ and $C$ with each a complexity of 1 and no interactions. $C_{SCM,A} = 2$ and $C_{SCM,B} = 1$.

2. Let $c$ be a non-negative number, and then there are only finite number of classes and programs of complexity $c$. *Proof*: Each class is composed of a finite number of components and thus each class has a minimum complexity before considering interactions. Thus, for any complexity $c$ there are a finite number of classes for which the sum of the complexity of components $\sum_{i=1}^{n} \alpha_i$ is less or equal to $c$ and the interaction complexity term $\left(\sum_{i=1}^{n} \sum_{j=1}^{n} \beta_{ij} + \sum_{i=1}^{n} \sum_{m=1}^{n} \gamma_{im}\right)\frac{v}{n}$ is smaller than $c - \sum_{i=1}^{n} \alpha_i$

3. There are distinct classes $P$ and $Q$ such that $|P| = |Q|$. This property states that there are multiple classes of the same complexity. *Proof:* Two different classes with no interactions may be composed of different sub-classes for which the sum of complexities is equal.

4. $(\exists P)(\exists Q)(P \equiv Q \ \&; \ |P| \neq |Q|)$. This property states that implementation is important. If there exist classes $P$ and $Q$ such that they produce the same output given the same input. *Proof*: Assuming that classes represent engineered systems, take two different that serve the same function, for example, to transport a person. An electric car, a gas car, or a bicycle could be used to transport a person, in each case, the "output", the transported person, is the same but the components, interactions, and architecture are different thus yielding different complexity results.

5. $(P)(Q)(|P| \leq |P;Q| \& |Q| \leq |P;Q|)$ This property states that if the combined class is constructed from class $P$ and class $Q$, the value of the class complexity for the combined class is larger than the value of the class complexity for the class $P$ or the class $Q$. *Proof*: Let set $A$ be a class of components $C$ and $D$. The complexity of $A$ is then

$$C_{SCM,A} = C_{SCM,C} + C_{SCM,D} + (\sum_{i=1}^{2} \sum_{j=1}^{2} \beta_{ij} + \sum_{i=1}^{2} \sum_{m=1}^{p} \gamma_{im})\frac{v}{2}$$

all values inequation in the previous equation are positive thus $C_{SCM,C} \leq C_{SCM,A}$ and $C_{SCM,D} \leq C_{SCM,A}$

6. $(\exists P)(\exists Q)(\exists R)(|P| = |Q|)\&(|P;R| \neq |Q;R|)\&(|R;P| \neq |R;Q|)$ This property states that if a new class is appended to two classes which have the same class complexity, the class complexities of two new combined classes are different or the interaction between $P$ and $R$ can be different than interaction between $Q$ and $R$ resulting in different complexity values for $|P;R|$ and $|Q;R|$. *Proof*: Assume $A$ and $B$ two classes of same complexity but with different components and interactions. Introduce a third interacting class $C$, the interactions between $A$ and $C$ and between $B$ and $C$ will be different due to the differences between $A$ and $B$ thus the complexity of the resulting class is also different.

7. There are program bodies $P$ and $Q$ such that $Q$ is formed by permuting the order of the statements of $P$, and $(|P| \neq |Q|)$. This property states that permutation of elements within the item being measured can change the metric values. The intent is to ensure that metric values change due to permutation of classes. *Proof*: If $Q$ is a permutation of the elements in $P$ then the interactions are also different resulting in a difference in the term $\sum_{i=1}^{n} \sum_{j=1}^{n} \beta_{ij}$ thus $(|P| \neq |Q|)$

8. If $P$ is renaming of $Q$, then $|P| = |Q||$. This property requires that when the name of the class or object changes it will not affect the complexity of the class. Even if the member function or member data name in the class change, the class complexity should remain unchanged. *Proof*: The complexity does not take the name of the components into consideration.

9. $(\exists P)(\exists Q)(|P| + |Q|) < (|P; Q|)$. This property states that the class complexity of a new class combined from two classes is greater than the sum of two individual class complexities. In other words, when two classes are combined, the interaction between classes can increase the complexity metric value. *Proof*: By definition, the complexity of the class is higher than the sum of the complexities of its components due to the added complexity due to the interactions.

### 2.3.2. Impact of Symmetry

The replacement of Graph Energy with cyclomatic complexity impacts the ability of the metric to capture the directionality of the interactions. Calculating both the SysML Complexity Metric and the DSM complexity metrics for all three cases shown in Figure 1 allows to see that the three cases have different topology but all three would have the same cyclomatic complexity. Taken as arbitrary values $\alpha_i = 1$ and $\beta_{ij} = 0.5$ the following results in Table 2 can be found.

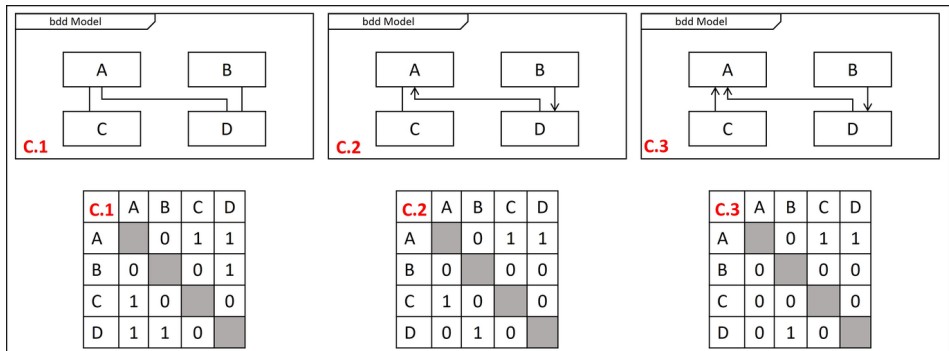

**Figure 1.** Example BDDs with the corresponding DSM Representations.

**Table 2.** Results on the impact of symmetry.

|        | $E(A)$ | $v$ | $C_{DSM}$ | $C_{SCM}$ |
|--------|--------|-----|-----------|-----------|
| Case 1 | 4.47   | 1   | 7.35      | 4.75      |
| Case 2 | 2      | 1   | 5         | 4.5       |
| Case 3 | 0      | 1   | 4         | 4.38      |

It is possible to see that as expected the graph energy varies depending on the symmetry but cyclomatic complexity does not, which results in symmetry having a bigger impact on the results for the DSM metric. It is to be noted that symmetry does impact the SCM based on the change in interactions $\beta_{ij}$. In summary, the DSM Complexity metric will be more sensitive to changes in symmetry than the SCM Complexity metric. Directed interactions are usually used to represent the direction of flow properties (energy, matter, and information). Given the definition of complexity chosen it is important to understand if a directed flow results in lower design, integration and operational effort. Depending on the impact of directionality, either the SCM or the DSM complexity might be more appropriate to assess complexity. For example, two components joined by a pipe need to have similar fittings on both ends, the effort to make this interface successful is not sensitive to the direction of the flow; in this case the direction of the interaction would be expected to have a low impact on the overall complexity. An information interface where there is a well-defined parent-child relationship where one component depends heavilyon the other might benefit from the stronger sensitivity.

With an initial understanding of the validity and behavior of the SCM complexity metric, it is now possible to investigate the applicability of the metric further by applying it to some case studies.

## 3. Case Studies

To evaluate the applicability, the proposed complexity metric will be applied to three different systems—a 3U CubeSat, a Mars lander, and a thermal management system within a spacecraft. The SCM metric will be then be calculated using simplifying assumptions. The results will then be compared to the DSM Complexity metric.

### 3.1. Case Study Methodology

All of the selected studies will follow the same methodology:

1. A type of system will be selected, and two variants of the system will be identified.
2. The two variants of a system will be decomposed into a set of parts.
3. Subsystems for each system will be identified and parts will be assigned to their corresponding subsystem.
4. A SysML representation of the system will be created with the structure and hierarchical decomposition shown in a BDD with a corresponding IBD showing the interactions.
5. The complexity of the components and the interactions will be determined.
6. The metric will be applied to the variants at the different decomposition levels, the subsystems levels, and the system level. Starting from the lowest level and using the information about the complexity of the level below to calculate the next level.
7. Sensitivity analyses will then be performed to determine the impact of the initial complexity assumptions.

Step 5 of the methodology requires a large quantity of information about the development of the system and its components. This information can take the form of quantifiable measures (time, cost, number of workers...) or the elicitation of expert opinion. These case studies aim to show the viability of the SysML metric rather than use the analysis for actionable decisions. Thus, step 5 will be performed using simplifying assumptions, such as assuming the same complexity for all components at the lowest decomposition level. If the metric is found to be viable further research will need to be performed to validate the metrics with the requisite quantifiable measures. In this paper, the metric will instead be validated by comparison with the closest established complexity estimation method in the body of knowledge: the DSM Complexity Metric.

To create a model from an existing DSM representation, the first step was to make the DSM symmetric by assuming any interaction to be bidirectional, a simplifying assumption given that it is not clear if a directional flow would reduce the interaction complexity. The parts were then divided into subsystems and an IBD was created for each subsystem depicting the interactions between the different parts of the subsystem. The interactions between parts of different subsystems were depicted as interactions between the subsystem themselves. This methodology renders the external interaction of the subsystems ($\gamma_{im}$) equal to zero. An illustration of the process can be seen in Figure 2.

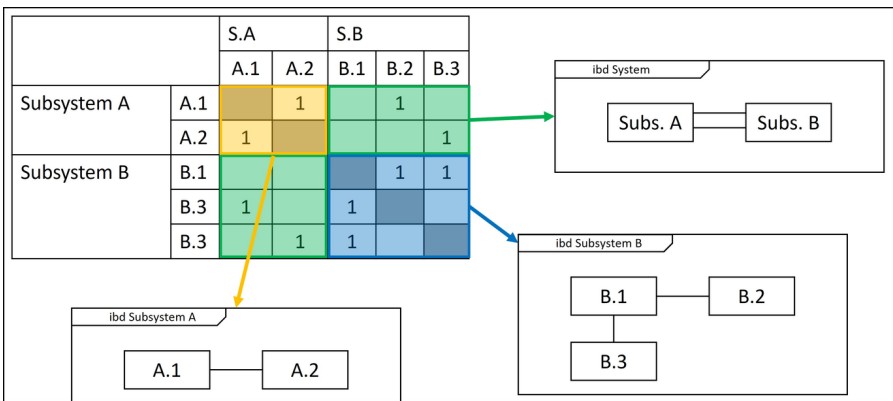

**Figure 2.** SysML model creation methodology.

*3.2. Mars Lander*

The first case study will look at two Mars landers for which the complexity has been assessed previously using DSMs [17]: The Mars Polar Lander (MPL) and Pathfinder. The Mars Polar Lander was launched on 3 January 1999 as a part of the Deep Space 2 Mission which had as an objective to explore the polar region of Mars for near-surface ice and to investigate the climate [17]. The lander mission failed just prior to cruise stage separation when all communications were lost. To this day, the reason for failure remains unconfirmed. Pathfinder, on the other hand, was launched on 4 December 1996, and successfully completed its mission of demonstrating a solution for placing a science payload on the surface of Mars [17]. The diagrams of both landers can be seen in Figure 3.

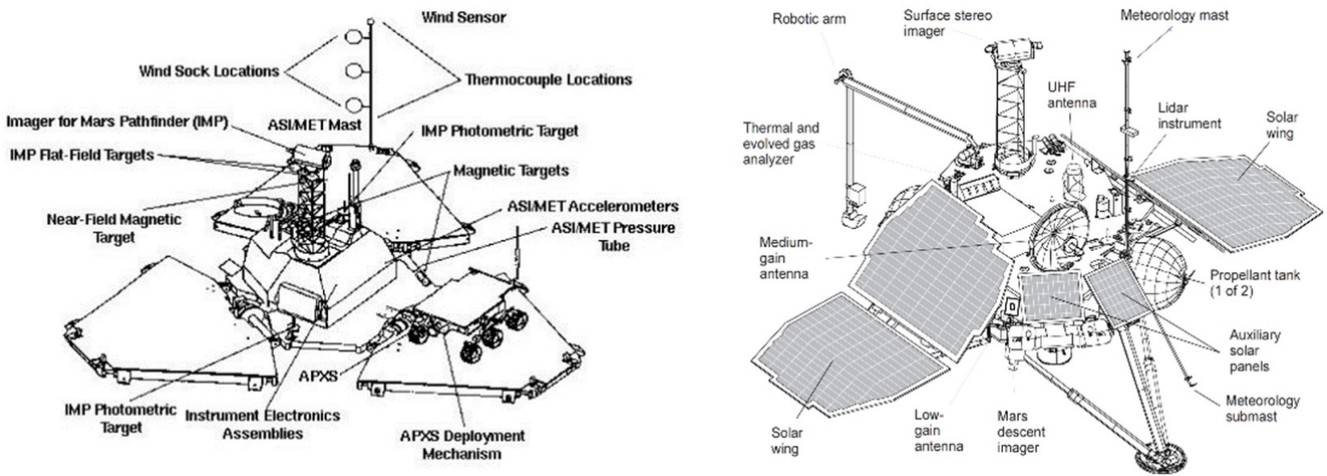

**Figure 3.** Mars Pathfinder Lander Configuration [17] (**left**). MarsPolar Lander diagram [18] (**right**).

The creation of the models for the analysis were realized using data from a DSM representation of the two variants of the system [17]. The Mars polar lander was decomposed into 32 components with 162 interactions and Pathfinder into 43 components with 154 interactions. Both systems were divided into the same set of seven subsystems as can be seen in Figure 4.

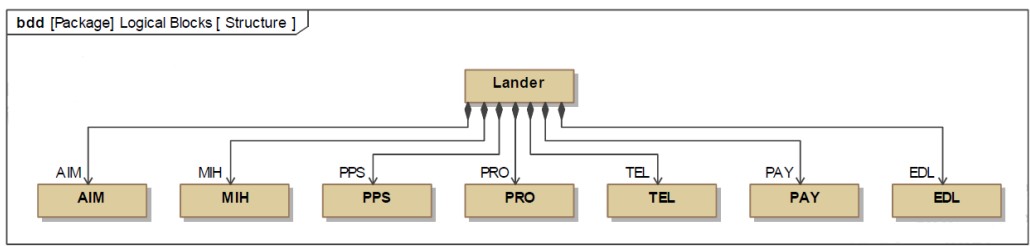

**Figure 4.** Lander Block Definition Diagram.

Each component was then assigned a subsystem and a set of BDDs was created for Pathfinder and MPL. An example of the BDD developed for Pathfinder can be found in Figure 5. Then following the methodology shown in Figure 2 associations between the different blocks were added.

Based on intuition alone it is hard to see which system is more complex assuming similar complexity for all components since the Pathfinder has more components, but the MPL has more interactions. Since unexpected failures are associated with complex systems it is then expected that between two comparable systems, a more complex one will be more likely to fail, thus given the historical failure of the MPL it is predicted that the MPL will yield a higher complexity value.

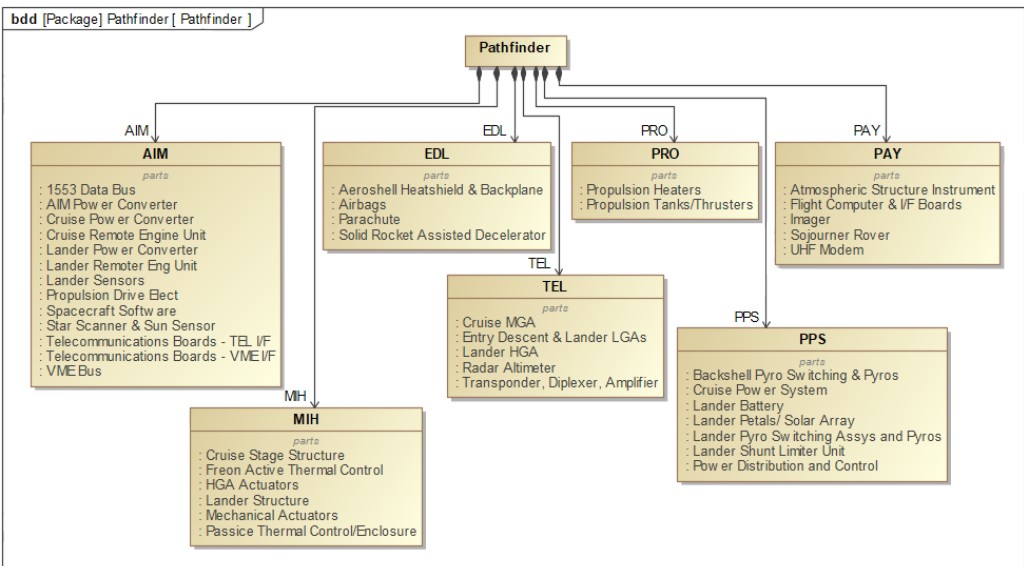

**Figure 5.** Pathfinder BDD.

The next step was to determine the complexity values of the components and the interactions. As an initial assumption the complexity value for all components at the lowest level of decomposition, $\alpha_i$ , was set to one. The rational for the assumption was that given the components were at the same level of decomposition, their complexity value should be similar. This assumption could be improved with more information about the resources needed to acquire each part such as cost or machining time, information that was not publicly available.

Next the assumption used by Sinha and Suh [19] to estimate the complexity of interactions to be equal to 10 percent of the mean of complexities of the two interacting objects was used for the lowest hierarchical level interactions; the complexity of the interactions within a subsystem, $\beta_{ij,subcomponent}$ , was set to be 0.1.

The complexity of the interaction between subsystems, $\beta_{ij,subsystem}$, was set to 0.2, double the complexity of interactions within a subsystem, based on the idea that the interaction between more complex components is more complex itself, the value would depend on the analysis context of development, for this study this choice will be retained as one to revisit in a later study if the metric is found to be viable.

With those assumptions, the complexity metric was applied to each subsystem. The metric was then applied to the lander as a whole using the susbystems as components. The results can be found in Table 3.

Overall, it can be observed that the MPL is considered 32.41% more complex than Pathfinder, even though the complexity of the sum of the components is higher for Pathfinder. This indicates that the more complex interactions of Pathfinder render the overall lander more complex, thus it would be expected that more effort and resources would be required to successfully develop MPL than Pathfinder.

To further the understanding of the impact of the component complexity assumption initially made, the analysis was reproduced while varying the component complexity, $\alpha_i$, between a value of one and ten, while retaining the assumption in Sinha and Suh of the interaction complexity, $\beta_{ij,subcomponent}$ being equal to be 10% of the component complexity, and the subsystem interaction complexity $\beta_{ij,subsystem}$ to be twice that value. This spread was selected since it provided a clear indication of the overall behavior of the metric with the changing parameters. The impact of such change on the overall system complexity can be seen in Figure 6. Furthermore, to understand the behavior of the metric with respect to the previous work realized using DSMs, the same analysis was reproduced using the DSM Complexity metric for comparison.

**Table 3.** Results of Lander Comparison.

| | MPL | | | Pathfinder | | | % Difference | | |
|---|---|---|---|---|---|---|---|---|---|
| | $\alpha_i$ | $\sum \beta_i$ | $v/n$ | $\alpha_i$ | $\sum \beta_i$ | $v/n$ | $\alpha_i$ | $\sum \beta_i$ | $v/n$ |
| AIM | 7.8 | 6.5 | 1.14 | 15.12 | 6.4 | 0.82 | 63.89 | 1.55 | 21.28 |
| EDL | 3.07 | 1.7 | 0.33 | 4.1 | 1.3 | 0.5 | 28.84 | 26.67 | 40 |
| MIH | 4.23 | 6.3 | 0.75 | 6.08 | 7.3 | 0.17 | 36.05 | 14.71 | 127.27 |
| PAY | 5 | 3.3 | 1 | 5.12 | 1.9 | 0.6 | 2.37 | 53.85 | 50 |
| PPS | 5.2 | 3.5 | 0.5 | 7.14 | 3.1 | 0.29 | 51.89 | 12.12 | 54.55 |
| PRO | 1 | 1.1 | 1 | 2.05 | 0.6 | 0.5 | 68.85 | 58.82 | 66.67 |
| TEL | 8 | 4.2 | 1 | 5.12 | 1.4 | 0.4 | 43.9 | 100 | 85.71 |
| $\sum \alpha_i$ (subs.) | 33.29 | | | 44.75 | | | 29.34 | | |
| $\sum \beta_i$ (subs.) | 26.6 | | | 22 | | | 18.93 | | |
| $v/n$ (subs.) | 18.29 | | | 15 | | | 19.74 | | |
| $C_{SCM}$ | 519.69 | | | 374.74 | | | 32.41 | | |

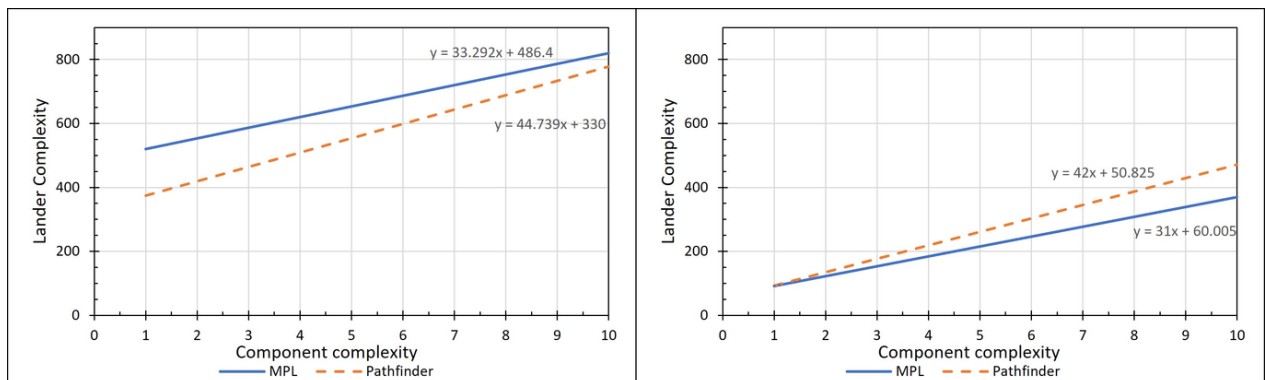

**Figure 6.** Impact of component complexity, $\alpha_i$, on the lander complexity with the SysML Complexity metric (**left**) and the DSM Complexity metric (**right**).

In Figure 7, it is possible to see that the complexity for Pathfinder rises faster than for the MPL which would indicate that Pathfinder was more sensitive to an increase in the complexity of its components $\alpha_i$. This is partly explained since Pathfinder has more components thus the complexity increase impacts more components.

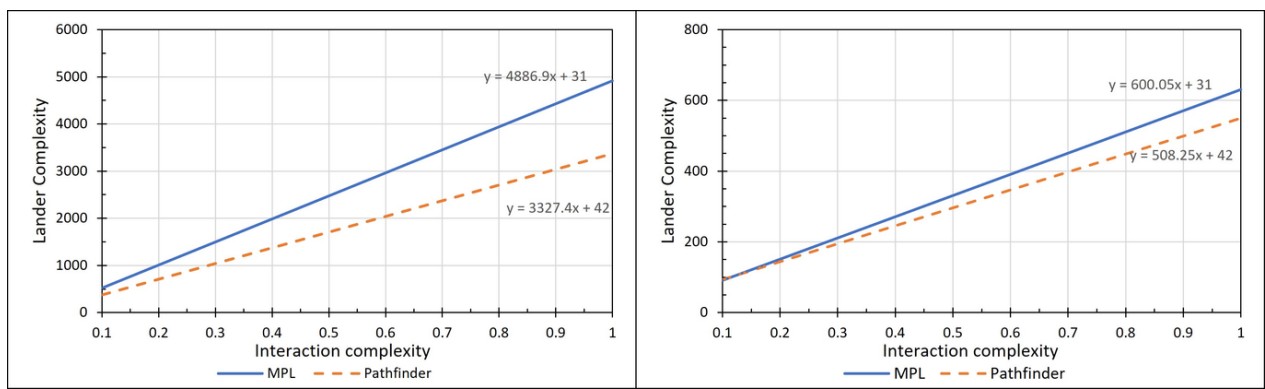

**Figure 7.** Impact of the interaction complexity value, $\beta_{ij}$, to the lander complexity with the SysML Complexity metric (**left**) and the DSM Complexity metric (**right**) (please note the difference in scale between the two plots).

The impact of the component interaction complexity, $\beta_{ij}$, was also investigated by varying the value of the interaction complexities, $\beta_{ij,subcomponent}$, from 0.1 to 1 while retaining the subsystem interaction complexities $\beta_{ij,subsystem}$, to be double the component interaction complexity. The results of this analysis can be found in Figure 7. Once again, to understand the behavior with respect to the previous metric the same analysis was reproduced using the DSM complexity metric.

Figure 7 shows that Pathfinder's complexity $C_{SCM}$ and $C_{DSM}$ rise at a lower rate than the MPL's complexity with both metrics, indicating that MPL is more sensitive to an increase in interaction complexity $\beta_{ij}$. This observation would translate that added difficulty in the interactions between the systems would result in higher development difficulty for the MPL compared to Pathfinder.

The use of the SysML Complexity metric revealed that the MPL appears to be more complex with the chosen initial assumptions, but it is noted that the opposite conclusion is found using the DSM Complexity metric. This result is because the DSM Complexity Metrics attributes less complexity due to the interactions than the SysML Complexity metric, and thus as can be seen in Figure 8 the intersection point between the lines is farther out. Nonetheless, the behavior of both metrics was similarly showing the MPL being more sensitive to an increase in interaction complexity $\beta_{ij}$, and Pathfinder to an increase in component complexity $\alpha_i$.

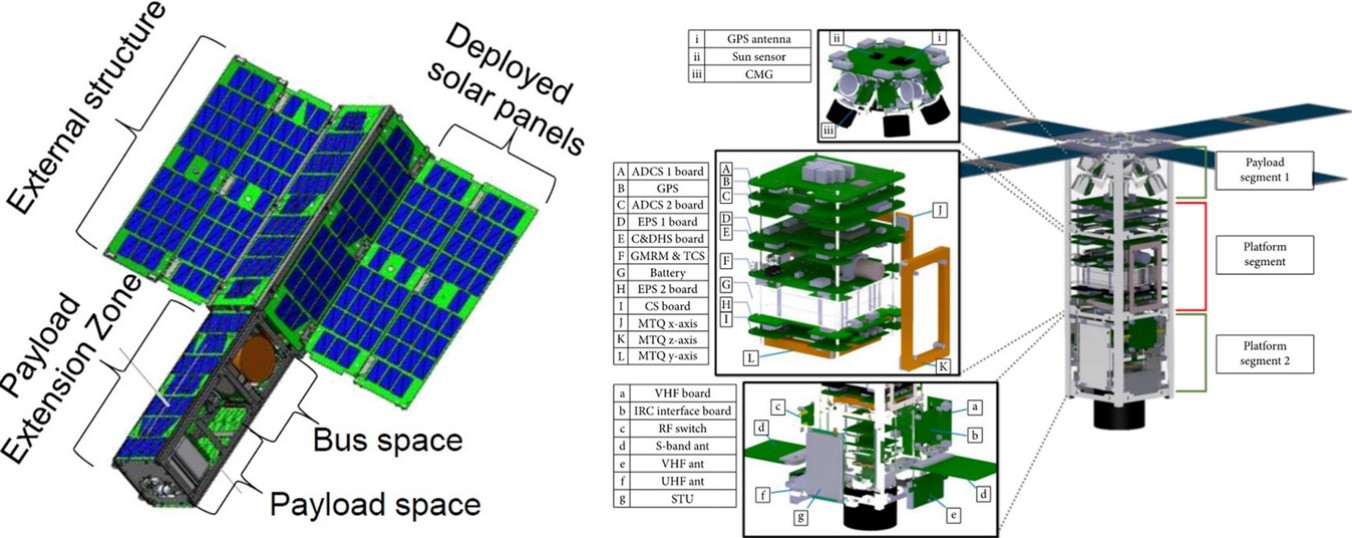

**Figure 8.** All-Star/THEIA CubeSat (**left**) and the KAUSAT 5 CubeSat (**right**).

With the initial assumptions, the MPL is found to be more complex than Pathfinder; additionally, it is more sensitive to an increase in interaction complexity. This added complexity would then translate into a higher effort needed to successfully integrate the MPL when compared to Pathfinder. It is then possible that with identical or similar resources and development procedures, the MPL development would encounter more issues with fewer resources to solve them. This could translate to additional verification tasks or tests needed that may have not been performed. This added complexity is not the reason for the actual failure of the system but may indicate the circumstances that could have made a failure more likely.

### 3.3. 3U Low Earth Orbit CubeSats

The second case study consists of two 3U flown CubeSats. The CubeSats were selected due to a detailed set of block diagrams publicly available. This case study shows how the metric can be applied with information that is already usually generated during the development of space systems. The first CubeSat is the All-Star/THEIA Mission a project of the Colorado Space Grant Consortium in collaboration with Lockheed Martin reported

by Patrick Blau [20]. The ALL-STAR/THEIA Mission launched on 18 April 2014 from Cape Canaveral, FL; Unfortunately, due to a malfunction on the ground station no data was recovered [21]. The second CubeSat the KAUSAT-5 3U CubeSat implemented and verified a standard platform architecture of 3U Cube Satellite proposed by the Song, Kim and Chang [22]. Both Cubesats are shown in Figure 8.

Both All-Star/THEIA and KAUSAT-5 are CubeSats adhering to the same platform 3U CubeSat platform with a similar intended low earth orbit as the operational environment. Based on the available documentations the systems were found to be composed of 66 parts with 125 interactions for All-Star/THEIA and 47 parts with 109 interactions for KAUSAT-5. Both CubeSats were decomposed into the same set of 6 subsystems with an additional specialized subsystem each. An overall DSM representation of components and interactions was created in order to be able to compare our results with the DSM Complexity metric. With the DSM created subsystems were represented into a set of BDDs and their respective IBDs as defined in Figure 2, similar to the Lander case study.

Similarly, to the Mars Lander's case studies, the complexity of all components, $\alpha_i$, will be assumed to be equal to one, and the complexity of all interactions between sub-components $\beta_{ij,subcomponent}$, will have a value of 0.1 and interactions between subsystems $\beta_{ij,subsystem}$, will have an interaction complexity of 0.2. With all the elements needed the metric was applied recursively to both CubeSats at both levels of decomposition. The results of the analysis can be seen in Table 4.

**Table 4.** Results of CubeSat Comparison.

| | All-Star/THEIA | | | KAUSAT-5 | | | % Difference | | |
|---|---|---|---|---|---|---|---|---|---|
| | $\alpha_i$ | $\sum \beta_i$ | $v/n$ | $\alpha_i$ | $\sum \beta_i$ | $v/n$ | $\alpha_i$ | $\sum \beta_i$ | $v/n$ |
| ACS | 27.11 | 0.85 | 0.96 | 10.27 | 1.00 | 0.30 | 90.11 | 16.22 | 104.50 |
| CDH | 16.92 | 0.70 | 0.44 | 7.09 | 0.70 | 0.14 | 81.93 | 0.00 | 101.54 |
| CS | 2.00 | 0.20 | 1.00 | 8.44 | 0.80 | 0.63 | 123.35 | 120.00 | 46.15 |
| EPS | 16.60 | 0.90 | 1.00 | 7.09 | 0.70 | 0.14 | 80.34 | 25.00 | 150.00 |
| PAY | 5.20 | 0.95 | 0.40 | 10.96 | 1.00 | 0.80 | 71.29 | 5.13 | 66.67 |
| PROP/TCS | 1.00 | 0.00 | 1.00 | 2.00 | 0.20 | 1.00 | 66.67 | 200.00 | 0.00 |
| STR | 6.40 | 0.30 | 1.40 | 3.20 | 0.30 | 0.67 | 66.67 | 0.00 | 70.97 |
| $\sum \alpha_i$ (subs) | 75.23 | | | 49.04 | | | 42.15 | | |
| $\sum \beta_i$ (subs) | 3.90 | | | 4.70 | | | 18.60 | | |
| $v/n$ (subs) | 3.57 | | | 8.71 | | | 83.72 | | |
| $C_{SCM}$ | 89.16 | | | 90.00 | | | 0.93 | | |

The complexity of AllStar/Theia $C_{SCM,AllStar/THEIA}$ is found to be equal to 89.16 to and the complexity of KAUSAT5 $C_{SCM,KAUSAT5}$ is found to be equal to 90.00. This results indicate that with the given assumptions the difficulty of acquiring either CubeSats to be very similar. In both cases the majority of the complexity originates mostly from the acquisition of individual components rather than from the integration of the spacecraft. As done previously, the complexity of the CubeSats was evaluated using the DSM complexity metric and the component complexity value, $\alpha_i$, was varied from 1 to 10 to investigate the impact of the assumption. The results of that analysis can be found in Figure 9.

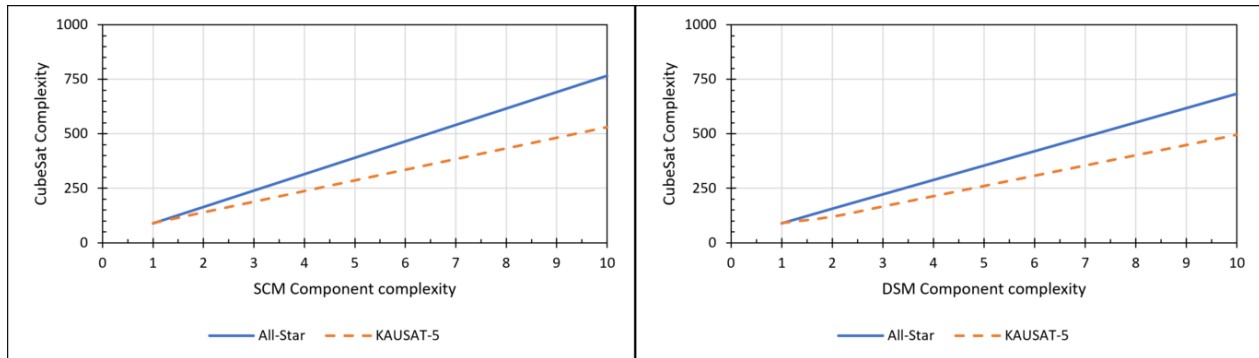

**Figure 9.** Impact of component complexity, $\alpha_i$, on cubesat complexity with the SysML Complexity metric (**left**) and the DSM Complexity metric (**right**).

Figure 9 shows that the All-Star/THEIA CubeSat is more sensitive to an increase in the component complexity, a trend that is also seen using the DSM complexity metric. In addition, the interaction complexity value, $\beta_{ij,subcomponent}$, was varied to investigate the impact of the assumption of $\beta_{ij}$. The analysis was reproduced with the DSM metric as well. The results of the analysis is found in Figure 10.

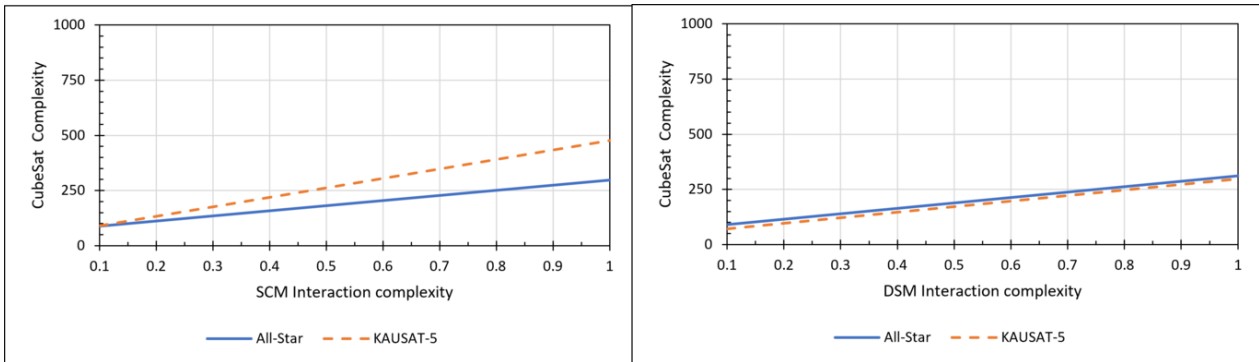

**Figure 10.** Impact of the interaction complexity value, $\beta_{ij}$, to the CubeSat complexity with the SysML Complexity metric (**left**) and the DSM Complexity metric (**right**).

Figure 10 shows that the complexity rises at a higher rate for the KAUSAT-5 than for the All-Star/THEIA mission. It is to be noted that even though the trend is the same using the DSM complexity metric the rate difference is a lot smaller making the lines appear almost parallel. This is due to fact that both the KAUSAT-5 and the All-Star/THEIA have similar $E(A)/n$ values, 1.429 and 1.489, respectively. With the KAUSAT-5 being more sensitive due to the higher number of cross-subsystem interactions.

### 3.4. Thermal Management System

The third study will explore how the assessment of complexity could be used as a factor during trade studies. With new spacecraft missions such as a manned flight to Mars, spacecraft with bigger propulsion systems and with bigger propellant tanks are needed. With this increase in propellant and tank size, propellant storage becomes challenging since those tanks are constantly receiving heat from the environment or other components of the spacecraft. A solution to this problem is to have a thermal management system that pumps heat out of the propellant tanks. The third case study will investigate two thermal management solutions. The starting point of this case study is a trade study by Plachta et al. [23] which studied different thermal management solutions for various tank sizes. The solutions considered are:

- A system using a single stage of cooling.
- A system using two stages of cooling.

Plachta et al. found that for some tank sizes a two stage cooling system results in an overall smaller mass. The objective is to provide system designers with additional insight by estimating the complexity impact of both solutions.

The first step in the study was to consider the scope of the complexity assessment. The scope will be similar to the one taken by Plachta et al. and only the thermal management system will be considered. The components considered for each solution can be seen in Figures 11 and 12.

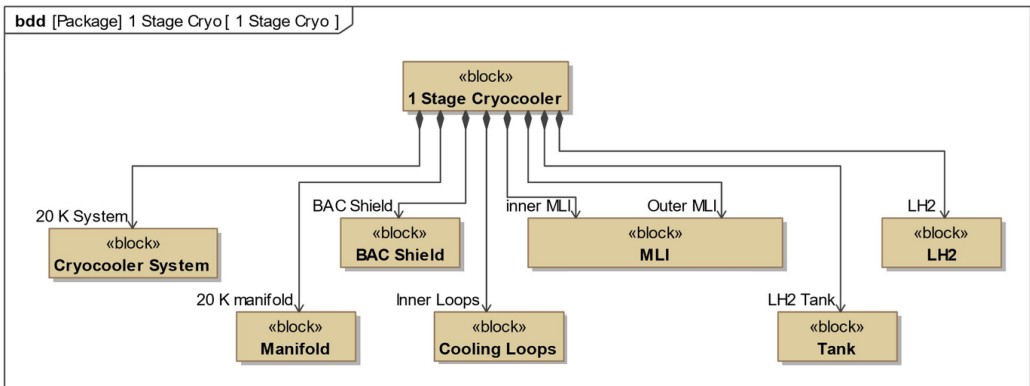

**Figure 11.** Single cooling circuit thermal management system BDD.

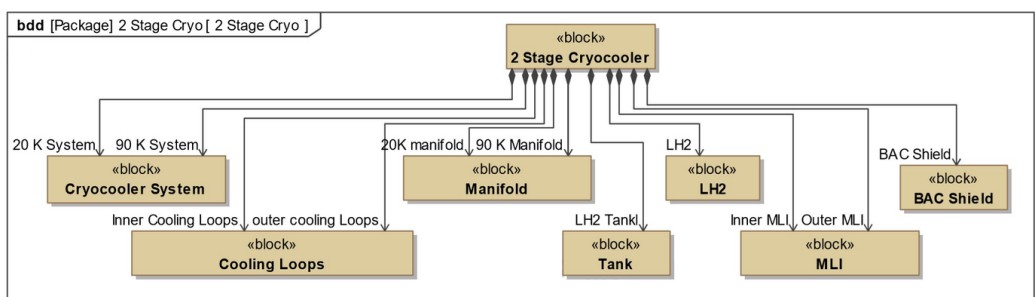

**Figure 12.** Two stage thermal management system BDD.

The next step in defining the scope is to determine the decomposition level needed for each component. In this case, all components in Figures 11 and 12 are considered simple enough thus do not warrant further decomposition except for the cryocooler system. In a real application, this decision would be made depending on the context of the analysis. For example, if the cryocooler system was going to be acquired from a third-party company, and that third-party company gave a specific time and cost for the system, then it would not be useful to decompose it further since it is possible to directly and reliably assess the effort required to acquire it and thus its complexity. In this case, it is assumed that the cryocooler systems would be developed alongside the rest of the thermal management system and thus the complexity of the cryocooler system is not known, warranting further decomposition seen in Figure 13. The ability to apply the metric at different scales for specific components allows for taking into account different systems of different scales in the same analysis.

Although the actual cryocooler systems may differ depending on the selected solution, an assumption will be made about all the cryocoolers having the same components and interactions shown in Figure 14.

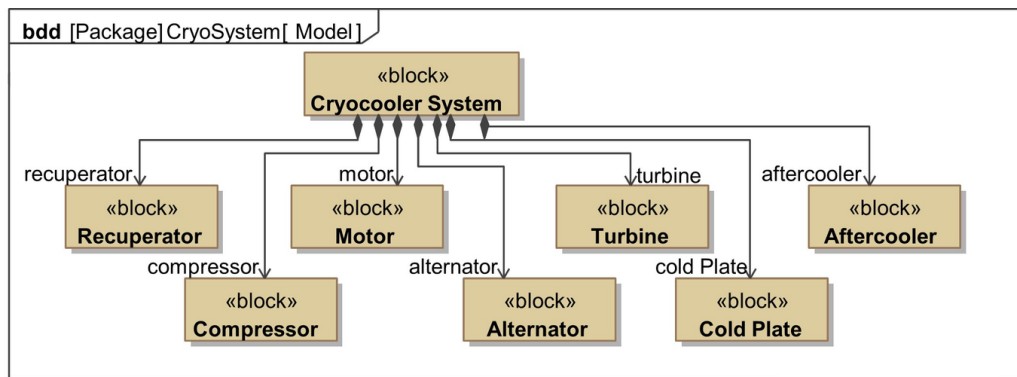

**Figure 13.** Cryocooler System BDD.

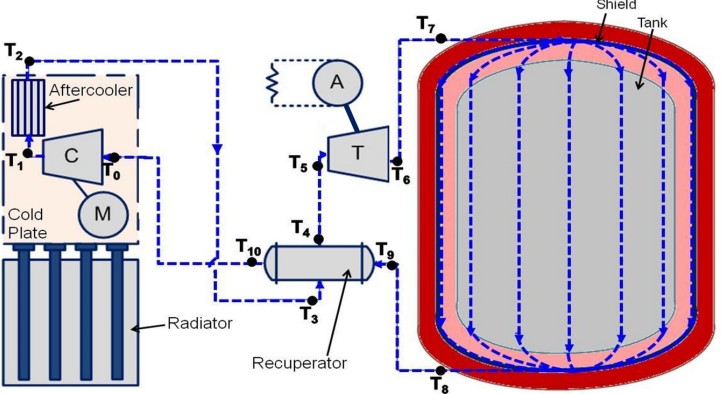

**Figure 14.** Cryocooler system schematic. Adapted from Guzik and Tomsik [24].

Using the information found in the analysis by Plachta et al. and in Guzik and Tomsik [23,24], IBDs representing the different flows between the components were created. The IBDs of the two stage and one stage thermal management system can be seen in Figures 15 and 16, respectively. Both systems show the different flows of working fluid and heat exchanges with the major difference being that the two stage system has an additional cryocooler system extracting heat from the outer MLI. All cryocooler systems are assumed to share the same architecture depicted in Figure 17.

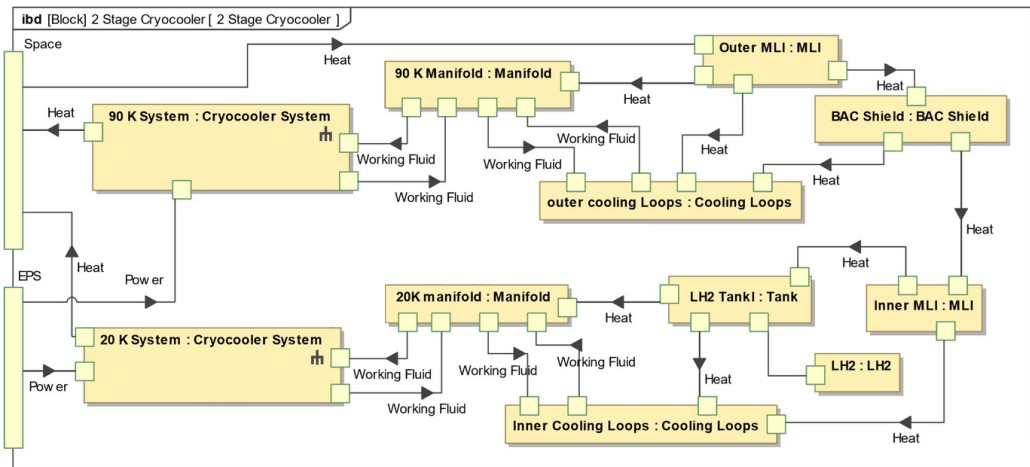

**Figure 15.** Two circuit cooling thermal management system IBD.

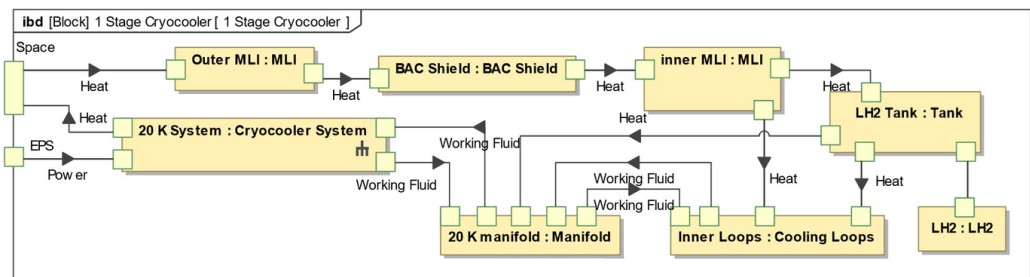

**Figure 16.** One stage thermal management system IBD.

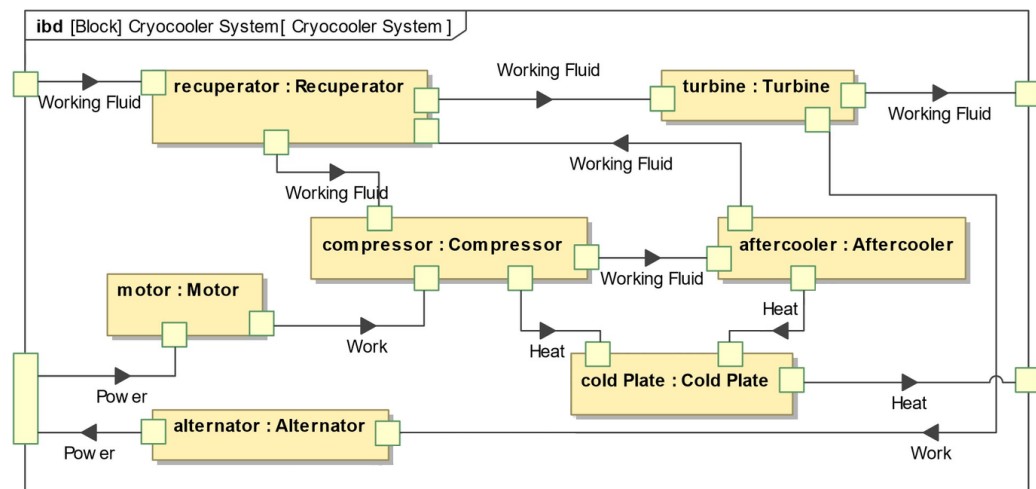

**Figure 17.** Cryocooler system BDD.

The assignment of complexity values, $\alpha_i$, to the components at the lowest levels of decomposition would be done based on previous similar system developments where comparable components were created or eliciting expert opinions on the matter. These complexity values would be dependent on the development context, the company resources, and expertise. Since the scope of this case study aims to be an illustrative process, showing the ability of the metric to provide insight, rather than an actual study used in the design of a spacecraft, basic assumptions were made about the complexity values assigned, making the complexity of the liquid hydrogen (LH2) a reference point with a value of $\alpha_{LH_2}$ equal to 1. It is to be noted that even though LH2 is not an engineered component, using the definition of complexity proposed it still requires resources to acquire and thus, has a complexity associated with it. A summary of the assumptions made, and their rationale can be found in Table 5.

With the complexity of the components $\alpha_i$ defined, it is necessary to assign a complexity value to each interaction, $\beta_{ij}$. Since in this case information about the type of interaction is available, specific complexity values will be assigned to the interactions based on their type rather than as a percentage of the complexity of the interacting components. The interaction complexity value needs to account for the effort required to make a successful connection between the components. For example, to establish a "working fluid" interaction, the effort would consist of performing the engineering analysis needed to determine the attributes relating to the interaction, and then the acquisition and installation of the physical apparatus needed, such as pipes and connections. Similarly, to the components, the assumptions for the complexity values for the different kinds of interactions and their rationale can be found in Table 6, where the interaction of the working fluid is determined to be the reference point with a value of 1.

**Table 5.** Component complexity values $\alpha_i$ and rationales.

| Component | Complexity | Rationale |
|---|---|---|
| Outer MLI | 3 | Sprayed MLI. Complex material installation |
| BAC | 2 | Similar to the tank complexity |
| Outer Cooling Loops | 2 | Simple materials, lines need to be attached to the tank |
| Inner MLI | 3 | Sprayed MLI. Complex material installation |
| LH2 | 1 | Reference Point. Engineering analysis needed. Acquired from a third party |
| LH2 Tank | 2 | Considered twice as complex as the LH2 itself since it needs engineering analysis and incur cost and installation effort. Acquired from a third party company |
| 20 K Manifold | 4 | Joint different tubes. Made in house/Simple materials but needed expert installation |
| Inner Cooling Loops | 2 | Lines/Simple materials needed to be attached to the tank |
| 90 K Manifold | 4 | Joint of different lines. Made in house/simple materials but requires expert installation |
| Cryocooler system component | 0.2 | Assumed to be 1/5 of the complexity of the LH2 tank. Acquired from 3rd party companies |

**Table 6.** Interaction complexity values $\beta_{ij}$ and $\gamma_{im}$ and rationales.

| Interaction Type | Complexity | Rationale |
|---|---|---|
| Working Fluid | 1 | Line connection assumed to be 1/4 the complexity of the manifold |
| Power | 0.4 | Need to select, acquire and install cables |
| Heat | 1 | Engineering analysis, no physical system to create |

All the information needed to assess the complexity of the cryocooler system, and the thermal management system are now defined. In an actual system development where MBSE is used, likely these diagrams would already exist and thus the complexity assessment would be easier. The diagrams created could also be used for other purposes; for example, the flow diagram could be used with a Modelica model integration to do a physical analysis of the system, or system constraints could be added and calculated using MATLAB. Furthermore, other systems engineering tasks such as requirement validation or communication could also use these diagrams. The results of the analysis using the initial assumptions can be found in Table 7. It is to be noted that since external interactions (the power from the spacecraft and the heat from space) are considered $\gamma_{im}$, it is not possible to apply the DSM Complexity metric as a comparison.

**Table 7.** Complexity assessment results for a thermal management system.

| | Cryocooler System | Single Stage of Cooling | Two Stages of Cooling |
|---|---|---|---|
| $\sum \alpha_i$ | 1.4 | 32.26 | 45.51 |
| $\beta_i$ | 9.5 | 5.2 | 10.1 |
| $v/n$ | 1.14 | 0.67 | 1.27 |
| $C_{SCM}$ | 12.26 | 35.72 | 58.37 |

In these results, it is possible to see that once again the resulting complexity $C_{SCM}$, of the cryocooler system is mostly due to the interactions rather than the sum of the components complexities, $\alpha_i$, since the sum of the components complexities is only equal to 1.40 against the cryocooler system complexity of $C_{SCM}$ equal to 12.26. It is also possible to see that using the initial assumptions, the thermal management system using two circuits of cooling is 48.14% more complex than the single stage of cooling system. This increase in complexity would then need to be accounted for when deciding between the two systems. In the work of Plachta et al. [23], it was seen that for large tank diameters (8.3 m and 6 m) the resulting mass for a two stage system is smaller than for one stage, even if slightly. With the additional information about the complexity of each solution, one would be able to include complexity as a factor in choosing between the single and dual loop architectures and make an overt decision whether the small reduction of mass is worth the growth in complexity.

To further understand the impact of the assumptions made, the analysis was reproduced where the 90 K cryocooler components were assigned a complexity of $\alpha_i$ equal to 0.02 and the 20 K cryocooler components a value of $\alpha_i$ equal 0.1. The rationale behind this is that adding a second cryocooler would alleviate some of the performance requirements of the 20 K cryocooler and the assumption that a 90 K cryocooler would have fewer complex components due to the higher control temperature.

In Table 8, it is possible to observe that with the assumptions made, the complexity of both cryocoolers is indeed smaller than the ones found previously. Even though the alleviation of the complexity values $\alpha_i$, the complexity $C_{SCM}$ of the two-stage cooling is still equal to 56.41 which represents only a 3.4% complexity reduction compared to the 58.37 found previously. This result indicates that even if the resulting cryocooler systems have significantly less complex components the overall thermal management system complexity remains a lot higher for the two-stage system. This result further indicates that the benefit from reduced performance requirements has little impact on the system as a whole and thus, the two-stage system remains more complex than the one stage system. The higher complexity measure $C_{SCM}$ indicates that more resources would be needed to develop the two-stage system. This fact could potentially raise a "red flag" to a system designer who might be misled into thinking that since the cryocooler is simpler, and that the overall mass is smaller, the two-stage solution will be easier to develop. Additionally, choosing the less complex solution would decrease the likelihood of encountering problems associated with complexity such as unexpected failures or schedule delays. This analysis method could be particularly useful in a situation where it is not just a single or two stage system being traded but a more complicated choice, for example between three or four cryocoolers where knowing which solution is more complex is not as intuitive.

**Table 8.** Complexity assessment result for a thermal management system with 90 K cryocooler component complexity of 0.02 and a 20 K Cryocooler component complexity of 0.1.

| | 20 K Cryocooler System | 90 K Cryocooler System | Single Stage of Cooling | Two Stages of Cooling |
|---|---|---|---|---|
| $\sum \alpha_i$ subs. | 0.7 | 0.14 | 31.56 | 43.55 |
| $\sum \beta_i$ subs. | 9.5 | 9.5 | 5.2 | 10.1 |
| $v/n$ (subs.) | 1.14 | 1.14 | 0.67 | 1.27 |
| $C_{SCM}$ | 11.56 | 11 | 35.02 | 56.41 |

## 4. Conclusions

### 4.1. Main Conclusions

The goal of this paper was to propose a metric that could provide system designers with quantitative insight into the complexity of their designs. This study established a working understanding of complexity based on real system properties in the context

of system development that can be understood for different systems of different scales. Using that understanding of complexity and previous work by Sinha et al. a structural complexity metric was proposed that leveraged the information about the composition and interactions of a system depicted in a SysML diagram. The metric was based on the DSM Complexity metric and was modified to allow for the metric to be applied at different system decomposition levels, be able to mix abstraction levels, provide the inclusion of external interactions, and be applied using a SysML diagram. The metric aims at solving two shortcomings of other metrics found in the literature by using a simple understanding of complexity related to effort of acquisition which could be approximated using available metrics such as cost and schedule and using SysML models that are commonly created for systems development thus reducing the effort required to implement and understand the metric. This metric was applied in three different case studies where it was used as a comparative analysis that provided additional insight into the different systems studied. The metric demonstrated that the modifications from the DSM complexity metric resulted in a higher amount of complexity attributed due to the interactions. This variance resulted in instances where conclusions about which system was more complex differed. Despite the difference in the allocation of complexity due to the interactions, both metrics behave similarly when increasing either the component complexity or interaction complexity thus showing that both metrics could be used to determine which system would be more greatly affected by an increase or decrease in the complexity of its components or interactions.

The SysML complexity metric could be added alongside other traditional project management techniques to offer system designers and project managers a new perspective on their design. The ability to integrate the metric at different levels for different components facilitates the use of the metric, particularly in projects that mix commercial of the shelve parts (COTS) and in house components. This new perspective should allow for a better understanding of the effort elicited by their design and thus should help designers better allocate their resources, avoiding needless complexity, especially in situations where it is hard to assess complexity based solely on intuition and validates the need for further investigations. In addition, as programs pursue a Model Based System Engineering approach, all the information required would already be developed during the development of the system, and the complexity estimation could be automated. This control on the system complexity should then result in better control of the problems associated with complexity such as budget overruns, schedule delays, or system failures.

Nonetheless, the SysML complexity metric does require the engineering system to be developed following a MBSE approach. In situations where the MBSE models do not exist, this approach would require significant time and effort. In addition, until further research is completed the SysML complexity metric does not escape the issues of the ambiguity of complexity, or expert opinion making the results subjective to the view on complexity and on the view of each expert. This ambiguity may hinder the credibility of the conclusions one may be able to gather from the application of the metric.

### 4.2. Future Work

The chosen case studies all relate to the field of space systems. To demonstrate the applicability of the metric to general systems, additional case studies with systems of different nature are envisioned.

The next step would then be to investigate the relationship between effort and resources spent in the specific domain of aerospace systems. This investigation should lead to an explicit association of effort to complexity values, where effort is a function of resources such as time and cost. Such an association would allow for an objective determination of the initial assumptions in the analysis ($\alpha$ and $\beta$) and verification of the metric. Using the initial assumptions, the behavior of the metric at different levels of decomposition could be studied to determine if the added complexity of the interactions is adequate, overestimated, or underestimated. With a calibrated metric, it would then be possible to inform design decisions such as developing a component in-house versus buying a

COTS option. Additionally, the metric could be used to optimize a system development to reduce overall system complexity by choosing different architectures, components, interactions, or by varying the allocation of resources available such as gaining more information through research or reducing the effort to make an interaction by hiring more experienced employees. This insight may uncover hidden dependencies allowing the improvement of the design process of aerospace system.

Additionally future work includes investigating other aspects of complexity including the dynamic aspects of complexity relating to the systems behavior. Instead of looking at the structural complexity of the system, the goal would be to evaluate the behavioral complexity. Behavioral complexity assessment could also leverage SysML models as these also usually contain behavioral expectations of the system. With both structural and behavioral complexity aspects studied an overall method could be proposed that takes into account both types of complexity.

**Author Contributions:** Conceptualization, V.E.P.L. and L.D.T.; methodology, V.E.P.L. and L.D.T.; software, V.E.P.L.; validation, V.E.P.L.; formal analysis, V.E.P.L.; investigation, V.E.P.L.; resources, V.E.P.L. and L.D.T.; data curation, V.E.P.L.; writing—original draft preparation, V.E.P.L.; writing—review and editing, V.E.P.L. and L.D.T.; visualization, V.E.P.L.; supervision, V.E.P.L. and L.D Thomas; project administration, V.E.P.L. and L.D.T.; funding acquisition, L.D.T. All authors have read and agreed to the published version of the manuscript.

**Funding:** This research received no external funding.

**Institutional Review Board Statement:** Not applicable.

**Informed Consent Statement:** Not applicable.

**Data Availability Statement:** Not applicable.

**Acknowledgments:** The writers wish to thank the faculty and staff of the University of Alabama in Huntsville's Department of Industrial & Systems Engineering and Engineering Management as well as the researchers at the Complex Systems Integration Laboratory (CSIL) for their support and valuable feedback.

**Conflicts of Interest:** The authors declare no conflict of interest.

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
