# Peer review of "Metric for Structural Complexity Assessment of Space Systems Modeled Using the System Modeling Language"

_aerospace, doi:10.3390/aerospace9100612_

Round 1

Reviewer 1 Report

Summary:

The paper proposed a new complexity metric, which is an adaptation of Sinha’s structural complexity framework. The adaptation and changes are supposed to make the metric suitable for the use with MBSE tools, such as SysML. The new metric is applied in three case studies and the results are discussed.
Feedback:
Positive: + the paper addresses a timely topic and uses the right/suitable approaches (e.g. Weyuker’s criteria) + the paper provides a comprehensive overview and the case studies add to the substantial nature of the work   Improvements need to be addressed: - the differences between the new metric and the one it is based on (Sinha) are not clearly outlined why they are advantageous and necessary - a general lack of comparison to recent literature and distinction from related approaches - the distinction regarding the assumptions made is not clear regarding the sours (Sinha); are the assumptions similar or the same? - lack of explanation of disadvantages/boundaries - format errors, missing captions (e.g. pg. 12), text extending into margins (e.g. pg. 4,7, ...), overlapping text (e.g. pg. 6) - very very wordy case study explanations with too much detail and too little focus on the results; the setups do not need to be explained in such an extensive way if not conducive to the results

Reviewer 2 Report

1. The first sentence of paragraph 2.2 "The metric shall rely on elements commonly accepted to contribute to structural complexity mentioned in the literature: (i) The size or number of parts 2 of a system (ii) the interactions between the parts and (iii) the intricacy of the interactions or modularity of the design.". I am not ready to agree with the authors that the modularity of the design contributes to an increase in the complexity of the product. On the contrary, the principles of modularity and unification are aimed at reducing the overall complexity.

2. One of the main elements of the proposed metric is the decomposition of the object of study into separate paths with an indication of the number of interactions between them. The question is to how the number of such interactions is calculated and how the authors propose to take into account the implicit (indirect) influences of some parts on others. For example, the thermal control system of a spacecraft can affect the operation of the receiving and transmitting equipment not directly, but through exposure the on-board computer or other elements of the satellite, etc.

3. As far as I understand, the authors (and not without reason) claim that their proposed metric allows you to make changes to the design process of an object based on the calculation of its complexity. This is an important conclusion that needs to be emphasized in the final part of the article and put this issue for future consideration, especially since the article contains a section "Future Work". It's a suggestion and recommendation but not obligatory.
